# Loss of c-Kit in Endothelial Cells Protects against Hindlimb Ischemia

**DOI:** 10.3390/biomedicines12061358

**Published:** 2024-06-19

**Authors:** Gustavo Falero-Diaz, Catarina de A. Barboza, Roberto I. Vazquez-Padron, Omaida C. Velazquez, Roberta M. Lassance-Soares

**Affiliations:** 1Department of Surgery, Miller School of Medicine, University of Miami, 1600 NW 10th Ave, RMSB, Miami, FL 33136, USA; gxf309@med.miami.edu (G.F.-D.); rvazquez@med.miami.edu (R.I.V.-P.); ovelazquez@med.miami.edu (O.C.V.); 2Department of Medicine, Miller School of Medicine, University of Miami, 1580 NW 10th Ave, Batchelor Building, Miami, FL 33136, USA; catarina.barboza@miami.edu

**Keywords:** c-Kit, SCF, arteriogenesis, endothelial barrier, CLI

## Abstract

Background: Critical limb ischemia (CLI) is the end stage of peripheral artery disease (PAD), and around 30% of CLI patients are ineligible for current treatments. The angiogenic benefits of c-Kit have been reported in the ischemia scenario; however, the present study demonstrates the effects of specific endothelial c-Kit signaling in arteriogenesis during hindlimb ischemia. Methods: We created conditional knockout mouse models that decrease c-Kit (c-Kit VE-Cadherin CreERT2—c-Kit) or its ligand (SCF VE-Cadherin CreERT2—SCF) specifically in endothelial cells (ECs) after tamoxifen treatment. These mice and a control group (wild-type VE-Cadherin CreERT2—WT) were subjected to hindlimb ischemia or aortic crush to evaluate perfusion/arteriogenesis and endothelial barrier permeability, respectively. Results: Our data confirmed the lower gene expression of c-Kit and SCF in the ECs of c-Kit and SCF mice, respectively. In addition, we confirmed the lower percentage of ECs positive for c-Kit in c-Kit mice. Further, we found that c-Kit and SCF mice had better limb perfusion and arteriogenesis compared to WT mice. We also demonstrated that c-Kit and SCF mice had a preserved endothelial barrier after aortic crush compared to WT. Conclusions: Our data demonstrate the deleterious effects of endothelial SCF/c-Kit signaling on arteriogenesis and endothelial barrier integrity.

## 1. Introduction

Peripheral arterial disease (PAD) is an occlusive disease that affects more than 200 million people worldwide [1,2], and approximately 50 million of these people are in Europe [3]. Critical limb ischemia (CLI) is the end stage of PAD; it is characterized by rest pain, gangrene, and resulting amputation. According to Fereydooni et al. (2019), the number of CLI patients in the US and Europe is very similar; in the US, it is approximately 2,595,676, while in Europe, it is about 2,551,917 [4]. The critical burden of CLI causes patients substantial risk and is associated with a fear of major amputation and a high mortality rate [5]. Moreover, the predicted 5-year mortality rates following a CLI diagnosis are as high as 50% [6], with most patients presenting with other serious vascular comorbidities, such as coronary artery diseases (CAD) and cerebral arterial disease [7,8]. Despite advances in diagnosis using state-of-the-art technology (such as diagnostic image exams), critical care, and surgical or endovascular revascularization, lower extremity amputation (LEA) due to PAD/CLI is still significantly high [9,10]. Moreover, a major proportion of CLI patients (20%) are unsuitable for revascularization interventions due to their high operative risk or unfavorable endovascular anatomy [11,12]. Thus, there is clearly an urgent unmet medical need to improve outcomes for these patients, and improving limb neovascularization is a promising approach. One of the two primary neovascularization processes is arteriogenesis, a process involving the growth and enlargement of pre-existing collaterals that can function as natural bypasses of a major occlusion [13,14]. This process, along with the hypoxia-mediated sprouting of a new capillary network (angiogenesis), ensures that sufficient blood flows to ischemic tissue after stenosis [15,16]. PAD patients who develop efficient arteriogenesis are less likely to progress to CLI [17]. Proper neovascularization requires endothelial barrier integrity and maintenance; otherwise, the collaterals cannot enlarge, and the capillaries will be dysfunctional leaking vessels.

c-Kit is a transmembrane receptor tyrosine kinase that, when activated by its ligand called stem cell factor (SCF), contributes to the growth, proliferation, migration, and differentiation of hematopoietic cells [18]. The -Kit signaling pathway has been extensively studied in the cancer field, and its angiogenic properties are well characterized [19,20]. Moreover, a few studies, including one from our group [21], have shown the beneficial effect of global c-Kit in improving arteriogenesis and limb perfusion during hindlimb ischemia [22,23]. However, the specific effect of vascular c-Kit signaling is poorly understood. Studies have shown that c-Kit plays a role in smoothing the muscle cells (SMCs), rescuing their contractile phenotype, and protecting against atherosclerosis. In addition, in endothelial cells (ECs), c-Kit increases endothelial barrier permeability [24,25]. These data together suggest a discrepancy in the role of c-Kit in the vasculature, having a protective effect in SMCs and a deleterious effect in ECs.

The present study assesses the consequences of targeting c-Kit/SCF signaling in ECs in the hindlimb ischemia scenario. c-Kit signaling has multiple and varied functions depending on the targeted disease and the cell type mechanism dissected. It is a pathway that has been studied in progenitor cells/hematopoiesis, cancer [26], atherosclerosis [27], angiogenesis [28], as well as arteriogenesis [21]. However, the specific role of endothelial c-Kit signaling in the hindlimb ischemia scenario has never been studied before. Here, we developed conditional knockout mouse models that, after TAM treatment, could specifically decrease the expression of c-Kit or SCF in ECs.

## 2. Methods

### 2.1. Animals

All animal experiments were approved by the University of Miami Miller School of Medicine Institutional Animal Care and Use Committee (IACUC) and followed NIH guidelines. For this study, we used 3 to 4-month-old males and females. We developed three mouse models to study the role of c-Kit signaling in endothelial cells, which are described below.

Endothelial c-Kit-deficient mice—c-Kit VE-Cadherin CreERT2: The Kit flox mouse (Kit^lox66−71/lox66−71^) from Dr. Rafii’s laboratory [29] (Weill Cornel Medical College) was bred with VE-Cadherin CreERT2 mice from Dr. Ralf Adams [30] [Cdh5(PAC)-CreERT2], which allowed Cre-specific recombination in ECs after TAM injections. This breeding strategy allowed us to generate the Kit conditional knockout c-Kit VE-Cadherin CreERT2 mouse (c-Kit). The experimental colony was established by crossing male c-Kit VE-Cadherin CreERT2 mice with female c-Kit VE-Cadherin CreERT2. Genotyping was performed by DNA PCR as previously described [29]. These animals were intraperitoneally injected with TAM (100 mg/kg body weight) for 1 week (to specifically inactivate the Kit gene in ECs) and rested the following week before being subjected to hindlimb ischemia or aortic crush procedures. The gene expression of c-Kit was examined in lung ECs from mice treated with TAM by qPCR; flow cytometry was also performed to analyze the percentage of endothelial cells that were also c-Kit positive.

Endothelial SCF deficient mice—SCF VE-Cadherin CreERT2: The SCF flox mouse (SCF^+/+^) obtained from Jackson Lab (stock# 017861) was bred with VE-Cadherin CreERT2 (donated by Dr. Dr. Ralf Adams mentioned above), which allowed Cre-specific recombination in ECs after TAM injections; as such, the SCF conditional knockout SCF VE-Cadherin CreERT2 mouse (SCF) was obtained. The experimental colony was established by crossing male SCF VE-Cadherin CreERT2 mice with female SCF VE-Cadherin CreERT2. Genotyping was performed by DNA PCR as mentioned above. These animals were treated with TAM, as mentioned above, to specifically inactivate the SCF gene in ECs and rested the following week before being subjected to hindlimb ischemia or aortic crush procedures. The gene expression of SCF was examined in lung ECs from mice treated with TAM by qPCR.

Endothelial CONTROL mice—WT VE-Cadherin CreERT2: The control mice (no flox gene) were bred with VE-Cadherin CreERT2, which allowed Cre-specific recombination in ECs after TAM injections. This produced the WT VE-Cadherin CreERT2 mouse (WT), where no genetic mutation occurred; however, it was possible to control the side effects of TAM. Genotyping and the TAM treatment were performed as described above. Even though these mice were treated with TAM to activate Cre recombinase, no genetic inactivation occurred; this is the appropriate mouse model to control for TAM side effects. The gene expression of SCF and c-Kit was examined in lung ECs treated with TAM by qPCR.

### 2.2. Hindlimb Ischemia

Femoral artery occlusion was performed as described. The mice were anesthetized with ketamine and xylazine, and their hindlimbs depilated. The left femoral artery was exposed through a 2 mm incision without retraction and with minimal tissue disturbance. A ligature was placed distal to the origin of the lateral caudal femoral and superficial epigastric arteries and proximal to the genu artery (below the inguinal ligament). The femoral artery was transected between the sutures and separated 1–2 mm. The wound was irrigated with saline, closed, and one dose of analgesia of buprenorphine slow release was administered [21].

### 2.3. Laser Doppler Imaging (LDI)

An LDI system (MoorLDI2-HR) was used to record the serial blood flow measurements before hindlimb ischemia, immediately post-operatively, and 3, 7, 14, and 21 days after hindlimb ischemia surgery. Excessive hair was removed from the limb before imaging, and the mice were placed on a heating pad at 37 °C to minimize temperature variation. The blood flow was measured and quantified in the foot of the animals, and the ratio between the ischemic and non-ischemic legs was determined to prevent variations in the body temperature of the mice interfering in the data quantification [21].

### 2.4. Immunohistochemistry (IHC)

Paraffin sections (5 μ thick) of the thigh were stained with smooth muscle actin (SMA), a marker of smooth muscle cells, to obtain the lumen diameter of the anterior and posterior gracilis muscle collaterals at the same level. The diameter of the collaterals was measured to quantify arteriogenesis. Briefly, after deparaffinization, antigens were retrieved from rehydrated sections using a Sodium citrate buffer (for SMA) for 20 min at 95 °C before treatment with 3% hydrogen peroxide and TNB blocking solution (cat. no. FP1020; PerkinElmer, Waltham, MA, USA). The slides were then incubated with Smooth Muscle Actin (SMA) antibody (Dako) overnight at 4 °C. The next day, the slides were incubated with anti-mouse biotinylated secondary antibody for 1 h, followed by streptavidin/HRP for 30 min. Detection was performed using a DAB kit, and slides were prepared by hematoxylin incubation and using mounting media with a coverslip.

### 2.5. Endothelial Barrier Injury (Aortic Crush Injury)

This procedure was carried out according to Yu et al. [31], with some modifications. The mouse models were divided into 6 different groups: WT aortic crush (WT AC), WT sham (WT S), SCF aortic crush (SCF AC), SCF sham (SCF S), c-Kit aortic crush (c-Kit AC), and c-Kit sham (c-Kit S). Briefly, animals under ketamine/xylazine anesthesia (100/10 mg/kg of body weight) were subjected to a median abdominal incision. The intestine was reflected laterally to the right, and a sterile gauze soaked with sterile saline was used to roll it up, mobilizing it to improve the exposure of the retroperitoneum and the abdominal aorta from the left renal vein to the aortic bifurcation. We used a sterile cotton-tipped applicator to deliver 30 consecutive crushes (each one of five seconds in duration) to the portion of the aorta from the renal vein to aortic bifurcation. Later, the viscera returned to their position and the fascia and skin were closed with sutures and postoperative care was performed. For the sham groups, the same surgical procedure was performed but without aortic crushes.

### 2.6. Evans Blue Quantification for Endothelial Barrier Integrity

Two days after the aortic crush or sham procedures, the mice were anesthetized (same dose mentioned above) for the endothelial barrier integrity analyses. They were perfused with 5 mL of 0.3% Evans Blue dye via the left ventricle, followed by 5 mL of PBS at a rate flow of 1 mL/minute. Next, the infrarenal aorta was dissected from adjacent tissues and excised from the left renal vein to the aortic bifurcation. The aorta was rinsed by passing it through PBS and put in 100 µL of 99% formamide solution for 4 h at 56 °C. All aortas were weighed, and the quantity of Evans Blue was quantified in each one using a standard curve of Evan Blue diluted in formamide. The optical densities of the samples and curves were obtained using a plate reader at 630 nm. All calculations were made by using the standard curve linear regression formula, and the quantity of Evan Blue per gram of aorta was recorded for further analysis.

### 2.7. Isolation of Lung ECs

The use of the lungs to extract endothelial cells was based on our experience, because a greater number of cells can be isolated from the lungs compared to the aorta. Animals were heparinized and anesthetized before the perfusion of 10–15 mL of PBS via the left ventricle. Both lungs were removed and placed in a cold medium. Under sterile conditions, the lungs were cut into small pieces and treated with collagenase–elastase (Worthintong Biomedical Corporation, Lakewood, NJ, USA, cat. number LK002066) at 1 mg/mL, in 15 mL of culture medium RPMI-1640 (Gibco, Life Technologies Corporation, Grand Island, NY, USA, cat. number A10491-01) per mouse, and in a shaker for 45 min at 37 °C. After that, the dissociated tissue was aspired by syringe with an 18G needle several times to triturate clumps and obtain a single-cell suspension. The cell suspension was passed through a 40 µm cell strainer on 15 mL of culture medium plus 10% fetal bovine serum to stop the digestion. Cells were counted and re-suspended in 0.5% BSA/PBS/EDTA. Finally, the lung endothelial cells were sorted using CD31 microbeads (Miltenyi Biotech, Bergisch Gladbach, Germany, cat. number 130-097-418) and magnetic separation following the manufacturer’s instructions.

### 2.8. qRT-PCR

After the isolation of lung ECs from all three mouse models (c-Kit VE-Cadherin CreERT2, SCF VE-Cadherin CreERT2, and WT VE-Cadherin CreERT2), total RNA was extracted using the E.Z.N.A Total RNA Kit from Omega Bio-tek (cat. no. R6834-02), and equal amounts of RNA were reverse transcribed using the High-Capacity cDNA Reverse Transcription Kit (Applied Biosystems by Thermo Fisher Scientific Baltics, UAB, Vilnius, Lithuania). The gene expression of SCF and KIT was measured by qRT-PCR, running in 20 μL reactions in a 7300 Real-time PCR system machine using TaqMan Gene Expression master mix (Applied Biosystems), according to the manufacturer’s instructions. Data were expressed as the relative fold change over the WT VE-Cadherin CreERT2 samples.

### 2.9. Flow Cytometry

The lung ECs were analyzed by flow cytometry. Briefly, one million cells were stained with BV421 rat anti-mouse CD31 Antibody (cat no. 562939) and BB115 rat anti-mouse CD117 Antibody, (cat. 564481), both from BD Biosciences. ECs were incubated with antibodies diluted 1/100 in FACS buffer (5% fetal bovine serum, 1 mM of EDTA, 0.1% sodium azide in phosphate-buffered saline) for 45 min on ice in the dark, then washed once, and fixed in FACS buffer plus 1% paraformaldehyde. Fixable Viability Dye eFluor 660 was used to irreversibly label dead cells prior to analysis. The labeled cells were analyzed using the BD LSRII from BD Biosciences. The LSRII is equipped with 4 lasers and 15 detectors and runs the FACS Diva v.8 acquisition software.

### 2.10. Data Analysis

Statistical calculations were performed using GraphPad Prism version 9.0.1 (San Diego, CA, USA). All values are expressed as mean ± standard error of mean (SEM). The data were tested for normality using either the Shapiro–Wilk test or QQ plot for samples that were too small. Welch’s *t* test was performed to compare the mean of the two groups with normally distributed data, and for the same comparison we used the nonparametric Mann–Whitney test for data that were not normally distributed. A one-sample *t* test was performed for a statistical comparison with the standard of the qPCR analyzed using the fold-change method. For the flow cytometry data, we used one-way analysis of variance (1-way ANOVA) with Dunnett’s posttest. An adjusted *p* value <0.05 was considered statistically significant.

## 3. Results

### 3.1. Confirmation of Animal Models

c-Kit is downregulated in the endothelial cells of c-Kit VE-Cadherin CreERT2. Cre mediates recombination in VE-Cadherin-positive cells in this model after TAM injections, which inactivate the c-Kit gene, specifically in ECs. The qPCR showed a significant reduction in c-Kit expression in lung ECs compared to WT VE-Cadherin CreERT2. In addition, and to confirm the specificity of the models, SCF VE-Cadherin CreERT2 did not show any significant difference in its c-Kit expression compared to WT VE-Cadherin CreERT2 (Figure 1A). To confirm the protein levels of c-Kit, we also submitted the lung ECs for flow cytometry analyses, which revealed a low percentage of CD31 cells (a marker of ECs) positive for c-Kit (Figure 1C). Together, these data confirm the efficacy of our mouse model, showing a decrease in endothelial c-Kit expression not only in gene expression, but also in protein levels.

SCF is downregulated in the endothelial cells of SCF VE-Cadherin CreERT2. The SCF model uses the same rationale as mentioned above, but the Cre mediates the inactivation of SCF in VE-Cadherin cells after TAM injections. Using qPCR, we showed a significant reduction in SCF expression in lung ECs compared to WT VE-Cadherin CreERT2. We also confirmed that c-Kit VE-Cadherin CreERT2 did not show any significant difference in SCF expression compared to WT VE-Cadherin CreERT2 (Figure 1B). Here, we also confirmed the efficacy and specificity of our mouse model.

### 3.2. Endothelial Barrier Function

#### Loss of Endothelial c-Kit Signaling Improves Endothelial Integrity after Aortic Crush Injury

The aortic crush injury data demonstrated the deleterious effects of c-Kit signaling on endothelial barrier integrity. Here, we subjected our three mouse models to the aortic crush injury or sham surgical procedures. We found that after aortic crush, the WT VE-Cadherin CreERT2, which had no SCF or c-Kit inactivation in the ECs (control), had greater levels of Evans Blue compared to the SCF VE-Cadherin CreERT2 and c-Kit VE-Cadherin CreERT2 mouse models. The protective effect of endothelial c-Kit signaling inactivation was remarkable, because the levels of Evans Blue after aortic crush on the SCF VE-Cadherin CreERT2 and c-Kit VE-Cadherin CreERT2 mouse models were similar to the Sham WT VE-Cadherin CreERT2 levels of Evans Blue (Figure 2). These findings suggested the deleterious effect of endothelial c-Kit signaling on endothelial barrier permeability.

### 3.3. Perfusion and Neovascularization

#### 3.3.1. Loss of Endothelial c-Kit Signaling Improves Blood Flow Recovery after Hindlimb Ischemia

Before ischemia, immediately after ischemia, and on days 3 and 7 after ischemia, there was no significant difference in the blood flow recovery between the groups. However, the inactivation of SCF (SCF VE-Cadherin CreERT2) or c-Kit (c-Kit VE-Cadherin CreERT2) in the endothelium showed a significant improvement in blood flow recovery on days 14 to 21 after hindlimb ischemia compared to the control (WT VE-Cadherin CreERT2) (Figure 3). These data together reveal the deleterious effect of endothelial c-Kit signaling on ischemic limb perfusion.

#### 3.3.2. Loss of Endothelial c-Kit Signaling Improves Arteriogenesis after Hindlimb Ischemia

The mean diameter of gracilis collateral vessels was quantified at 28 days after hindlimb ischemia by the immunohistochemical staining of SMA. As shown in Figure 4, while the control group (WT VE-Cadherin CreERT2) displayed baseline collateral remodeling, mice with the loss of endothelial c-Kit (c-Kit VE-Cadherin CreERT2) showed a trend of improved arteriogenesis and mice with the loss of endothelial SCF (SCF VE-Cadherin CreERT2) showed significant arteriogenesis.

## 4. Discussion

PAD and CLI are major economic burdens to the health systems of many countries. Although this burden is difficult to predict because most studies only include hospitalized patients, a recent study estimated that the annual Medicare cost of CLI is USD 12 billion in the US alone [32]. In Europe, specifically in countries like the Netherlands, the cost per PAD patient is about EUR 17 thousand [33], while in Germany, the total cost of early endovascular intervention is approximately EUR 24 thousand per patient over 5 years [34]. Despite advances in diagnostic imaging, including magnetic resonance angiography, intravascular ultrasound, and optical coherence tomography [35], the PAD/CLI scenario is still unfavorable for many patients, and there is a real need for novel therapeutic approaches that can treat this critical unmet medical need. Thus, the present study reveals the c-Kit signaling pathway in endothelial cells as a potential therapeutic target for future treatments in PAD/CLI. Our study is the first to demonstrate that endothelial c-Kit signaling is deleterious during hindlimb ischemia. We found that the loss of c-Kit signaling in ECs improves perfusion and arteriogenesis after limb ischemia. In addition, we revealed that the loss of endothelial c-Kit or SCF protects the integrity of the endothelial barrier after aorta damage. Our findings suggest that targeting endothelial SCF/c-Kit signaling could improve the outcomes in the limb ischemia scenario.

Several reports have demonstrated the effect of global c-Kit signaling in the hindlimb ischemia scenario. The association between endothelial progenitor cell mobilization (EPCs—most of them c-Kit+/CD31+ cells derived from the bone marrow), limb ischemia and an improvement in perfusion and neovascularization has been well documented [36,37]. In addition, the delivery of transmembrane SCF in a rabbit hindlimb ischemia model showed an improvement in revascularization, as well as a greater number of vessels in the ischemic foot [38]; meanwhile, the loss of global c-Kit (using a c-Kit mutant mice), shown by our group and others, impairs limb perfusion, arteriogenesis, and limb function in the mouse model of hindlimb ischemia [21,36]. We believe that the discrepancy between a global mutation of c-Kit showing a negative effect in arteriogenesis and a specific EC c-Kit inactivation revealing a positive effect in arteriogenesis (both studies using the same model of limb ischemia) is related to the specificity of c-Kit inactivation in the endothelial cells in the present study. c-Kit signaling might contribute significantly to arteriogenesis in other types of cells, such as SMCs, immune cells and progenitor cells; therefore, when this signaling pathway is lost globally (including endothelial cells), the negative effect of the loss of c-Kit in all other cell types overcomes the positive effect of c-Kit loss in endothelial cells. The literature on c-Kit signaling and hindlimb ischemia has always been based on a global investigation of this pathway, without looking specifically at the role of this pathway in vascular cells.

The role of c-Kit signaling in other major ischemic diseases, such as myocardium infarction (MI) and stroke, has also been studied. These findings revealed the ability of bone marrow-derived c-Kit positive cells, as well as SCF administration, to improve the outcomes of MI [39]; interestingly, a novel c-Kit ligand called meteorin-like also improved the deleterious effects of MI by increasing capillary formation in the MI border zone [40]. Regarding stroke, the inhibition of microRNA-124 has been found to activate SCF/c-Kit signaling, leading to blood vessel formation [41], while SCF treatment has been found to promote neurogenesis in rats [42].

The angiogenic effects of c-Kit signaling during ischemia [43], retina pathological neovascularization [28], and in the cancer field [44] are well described in the literature. Although neovascularization may not be a welcome process in the retina and in cancer, c-Kit signaling has always been seen as contributing positively to increasing angiogenesis for ischemia diseases. Of note, angiogenesis is a totally different process compared to arteriogenesis; the latter is triggered by hemodynamic changes and the enlargement/remodeling of tiny pre-existing vessels that are already established in the vasculature [45]. Moreover, most of the existing studies on ischemia, especially in peripheral limb ischemia, do not identify the role of this signaling pathway in the endothelium, due to the lack of in vivo tools, such as the transgenic models presented in this study. Several reports have assessed c-Kit signaling using in vitro angiogenesis by targeting c-Kit and its ligand SCF in endothelial cells [46,47,48,49]. Today, however, cell-specific transgenic mouse models provide important, widely used tools that allow the investigation of a specific gene in a specific cell line [50]. Specific conditional transgenic mouse models, in our case the loss-of-function of c-Kit or SCF specifically in endothelial cells, allow us to inactivate/decrease the expression of these genes using TAM treatment when these mice are adults, avoiding vascular development/morphology compromise during the embryonic phase, especially because of the role of c-Kit during embryonic vascular development [51,52]. Moreover, the use of a WT transgenic mouse model carrying Cre recombinase (WT VE-Cadherin CreERT2) is an appropriate control because it allows the impact of the TAM treatment to be controlled.

Our group has also published research on another conditional transgenic mouse model that inactivates c-Kit in smooth muscle cells to assess this pathway during atherosclerosis. This research concluded that c-Kit in SMCs protects against atherosclerosis in ApoE KO mice fed with a high-fat diet [53]. Interestingly, while we observed a protective effect of c-Kit in SMCs during atherosclerosis, the present study shows the deleterious effects of the same pathway, suggesting that the cell line and the type of disease are key factors that need to be considered during the study of a specific pathway. In other words, c-Kit signaling appears to have several roles, specifically depending on the type of cell and the stimulus the cells are exposed to. Our group has investigated the protective effect of c-Kit in smooth muscle cells during atherosclerosis [27,53] on the nitric oxide vasodilation pathway [54], on the prevention of the SMC pro-inflammatory phenotype [27,55] and during arteriogenesis [21].

Even though most of these reports show the beneficial effects of c-Kit, other studies have revealed that c-Kit plays the role of villain when in endothelial cells, inducing pathological neovascularization in the retina [56], damaging glomerular endothelial cells during diabetic nephropathy [57], increasing vascular permeability and leakage vessels [24,25], and leading to blood–brain barrier permeability through endothelial dysfunction during chronic cerebral hypoperfusion [58].

The endothelial barrier is essential for homeostasis by controlling the passage of macromolecules, immune cells, and solutes from the plasma to the surrounding tissue and maintaining the plasma volume [42,59]. Vascular homeostasis is also essential during arteriogenesis; for proper collateral remodeling, a limited inflammatory process with controlled vascular cell proliferation is key; however, if the endothelial barrier is destabilized, leukocyte extravasation is enhanced, leading to a significant inflammatory process. This might be a reasonable mechanism to explain why the loss of c-Kit in endothelial cells protects the endothelial barrier, limiting leukocyte extravasation/inflammation, which is beneficial for proper arteriogenesis. However, more studies are needed to dissect in detail the mechanisms by which c-Kit impairs the function of the endothelial barrier.

In summary, this study is highly significant because it is the first to show the deleterious effect of endothelial c-Kit signaling on arteriogenesis after hindlimb ischemia, which is potentially associated with an impaired endothelial barrier. Although further investigation is required to delineate the molecular mechanisms underlying endothelial c-Kit signaling during arteriogenesis, our study reveals for the first time the importance of cell type specificity in this pathway and opens new avenues showing endothelial c-Kit as a potential therapeutic target for PAD/CLI in the future.

Limitation: The present study demonstrated the role of endothelial c-Kit signaling in arteriogenesis using a specific Cre conditional knockout model. However, one of our major limitations here is that the loss of c-Kit (gene and protein levels) in the endothelial cell is not complete; it is reduced by about 50%. However, it still shows a significant effect on hindlimb ischemia and endothelial permeability. Moreover, even though we demonstrated convincing data about the deleterious effects of endothelial c-Kit signaling on limb perfusion and arteriogenesis, we used a mouse model of hindlimb ischemia, so future studies are needed to translate it to humans. Although we pointed out important limitations here, we believe this study can open new therapeutic avenues targeting endothelial c-Kit signaling to improve the outcomes of PAD/CLI.

## Figures and Tables

**Figure 1 biomedicines-12-01358-f001:**
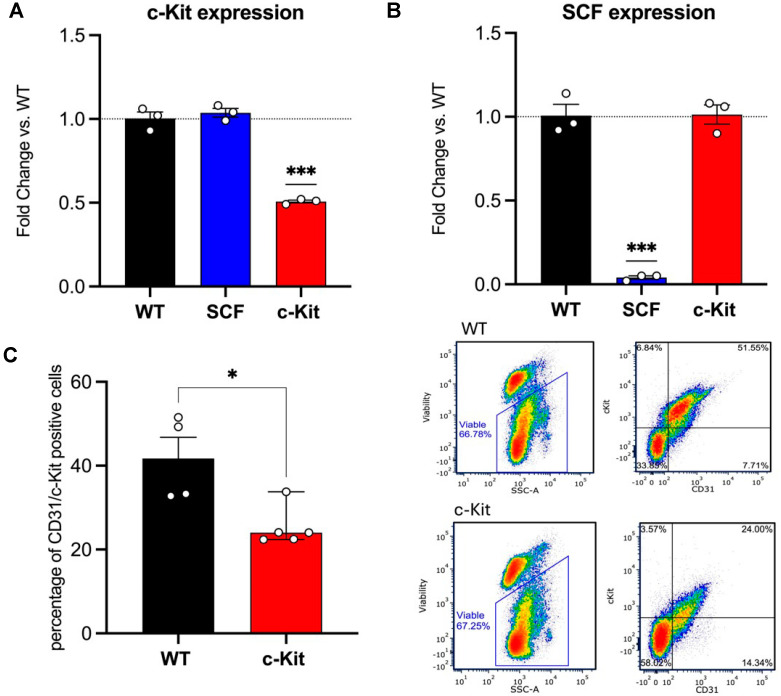
c-Kit and SCF expression. (**A**) c-Kit VE-Cadherin CreERT2 mice showed a significant decrease in c-Kit gene expression compared to WT VE-Cadherin CreERT2 and SCF VE-Cadherin CreERT2 mice in lung ECs tested by qPCR. (**B**) SCF VE-Cadherin CreERT2 mice showed a significant decrease in SCF gene expression compared to WT VE–Cadherin CreERT2 and c-Kit VECadherin CreERT2 mice in lung ECs using qPCR. (**C**) c-Kit VE-Cadherin CreERT2 mice showed fewer ECs (CD31+ cells) positive for c-Kit compared to WT VE-Cadherin CreERT2 mice using flow cytometry analysis. * *p* > 0.05; *** *p* > 0.001.

**Figure 2 biomedicines-12-01358-f002:**
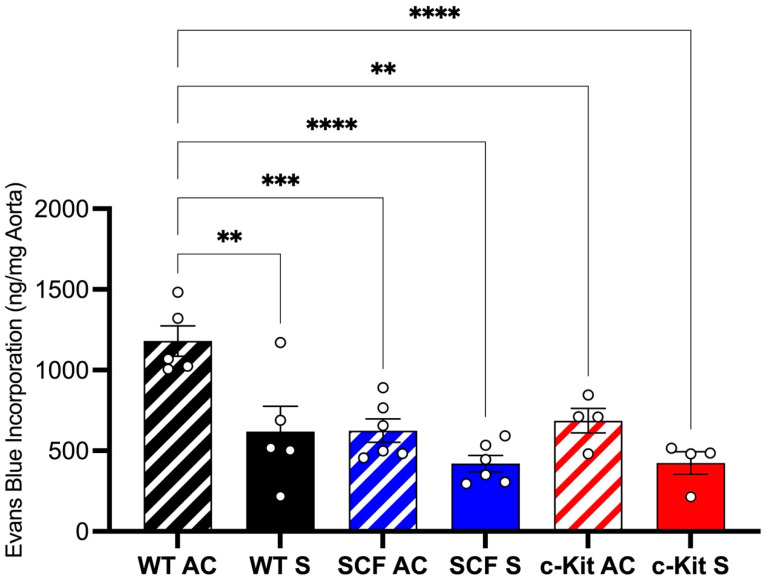
Loss of endothelial c-Kit signaling protects against endothelial barrier damage. cKit VE-Cadherin CreERT2 as well as SCF VE-Cadherin CreERT2 mice showed lower Evans Blue incorporation after aortic crush procedure. Of note, the c-Kit VE-Cadherin CreERT2 and SCF VECadherin CreERT2 aortic crush groups are similar to the WT VE-Cadherin CreERT2 sham group, suggesting that loss of this pathway in endothelial cells protects against endothelial permeability. AC, aortic crush; S, sham. ** *p* > 0.01; *** *p* > 0.001; **** *p*>0.0001.

**Figure 3 biomedicines-12-01358-f003:**
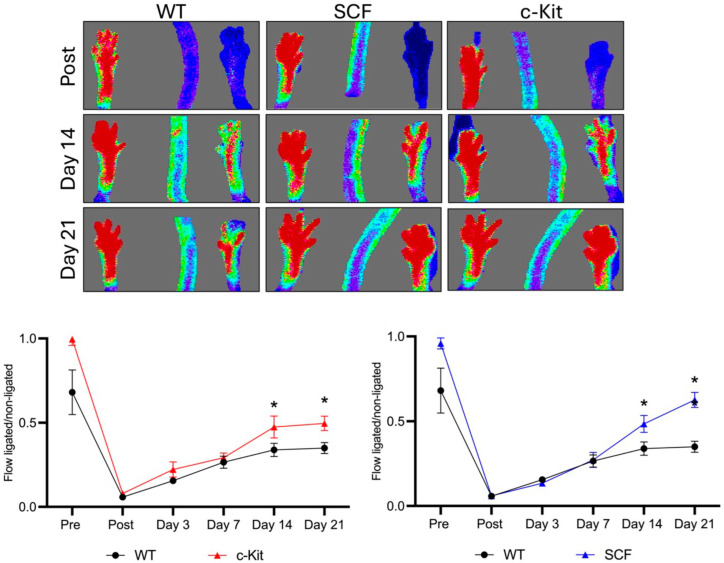
Loss of endothelial c-Kit signaling improves ischemic limb perfusion. LDI demonstrates a significant improvement in blood flow recovery on days 14 and 21 in c-Kit VE-Cadherin CreERT2 and SCF VE-Cadherin CreERT2 compared to WT VECadherin CreERT2. * *p* > 0.05.

**Figure 4 biomedicines-12-01358-f004:**
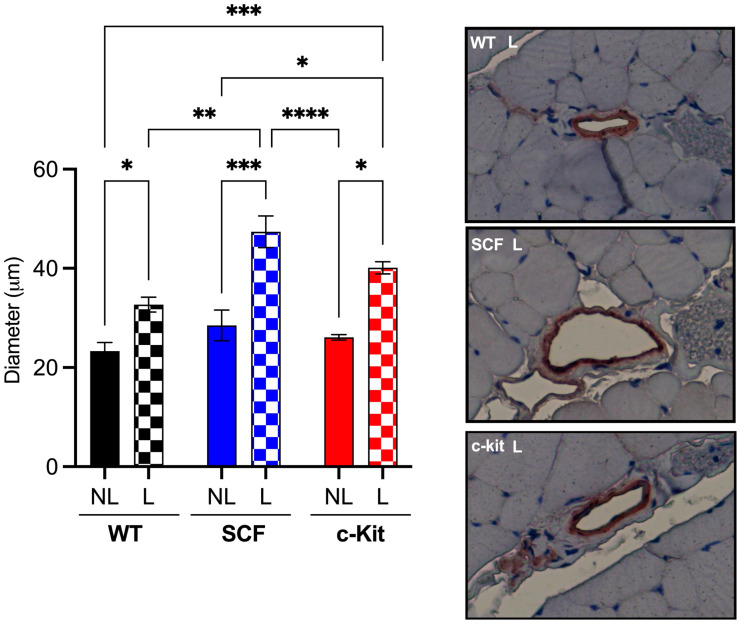
Loss of endothelial c-Kit signaling improves arteriogenesis after hindlimb ischemia. These data together strongly suggest that endothelial c-Kit signaling has a significant deleterious impact on perfusion due to poor arteriogenesis after hindlimb ischemia. * *p* > 0.05; ** *p* > 0.01; *** *p* > 0.001; **** *p* > 0.0001. Magnification: 40×.

## Data Availability

Data will be made available upon request.

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
