# Peer review of "Loss of c-Kit in Endothelial Cells Protects against Hindlimb Ischemia"

_biomedicines, 2024, doi:10.3390/biomedicines12061358_

Round 1

Reviewer 1 Report

Comments and Suggestions for Authors

Reviewer 2 Report

Comments and Suggestions for Authors

The present study aimed to assess the consequences of targeting c-Kit/SCF signaling in ECs in the hindlimb ischemia scenario.

Although it is an interesting subject, the manuscript has some points that should be addressed before being considered for publication.

1)      The purpose of the study should be clear at the end of the introduction session. The authors opted to write the conclusion instead of the purpose of the study;

2)      The reason/justification of the study is much better explained in the first paragraphs of the discussion session. I suggest bringing it to the introduction session;

3)      Line 49-50: “Moreover, few studies, including one from our group [14] have shown the beneficial effect 49 of c-Kit in improving arteriogenesis and limb perfusion during hindlimb ischemia[15,16]”. On the other hand, the conclusion of the study demonstrated that the loss of c-Kit protects against hindlimb ischemia in endothelial cells. It doesn't seem very clear. Could you explain the primary function of c-Kit in the endothelial cell ?

4)      All results could be described in more detail. For example, Figure 1C demonstrated an interesting result, however, the results are explained in a straightforward manner.

5)      The figure 3 demonstrated an interesting result (especially in days 14 and 21). However, how can one ensure that other systems, such nitric oxide, aren't influencing this outcome?

Round 2

Reviewer 1 Report

Comments and Suggestions for Authors

Dear Authors,

I believe that your findings are clinically important and valuable. But somehow, the meaning of your research is escaping the attention. You have made elaborated and contemporary research evolving modern laboratory tests and current up-to-date methods. However, you have put your results against old epidemiological data. Although the papers by Allison (ref 1) and Hirsch (ref 2) concern the epidemiological issue, they are simply out-of-date as their findings relate to PAOD epidemiology in mid-XX Century. In the Western world, lower extremity amputation (LEA) most commonly results from critical complications of peripheral arterial occlusive disease (PAOD) or diabetic foot disease. Despite huge progress in Medicine (diagnostic imaging tools work-ups like CTA and MRA, IVUS, OCT, as well as modern antidiabetic medications),  the LEA incidence in the USA and European Union countries for the years 1990-2017 varies accross different countries. E.g. the Estimated Annual Percentage Change is higher in UK, Sweden,  and USA in years 2009-2017 compared to former time periods (
https://doi.org/10.1016/j.ejvs.2020.05.037). Only, in Poland there are 40,000 hospitalizations for PAOD, and 9000 limb amputations per year, which means that limb salvage projects are an illusion rather than a real action (https://doi.org/10.3390/jcm13051471).

Therefore in this context, your findings are extremely important, but you should emphasize the scale of the disease together with the potential therapeutic solutions that result from your research.  To do so, I would advice to show up-to-date epidemiology. The cited references from 2005 (although summarizing data from XX century) are simply inadequate. The US and EU data are refreshed, due to the progress made in the diagnostic work-ups, including OCT, IVUS, computed tomography and magnetic resonance angiography, that allow better assessment of PAOD presence and extent. Furthermore, a constant efforts made in the improvement of medications that control the disease, result in RCTs that contribute to the reduction in the adverse limb ischemia events. 

Major comments: as above. The introduction and discussion needs up-to-date epidemiological data and a paragraph on the novel diagnostic tools and medical interventions in the context of the Authors research findings.

Minor comments: PIMD (PMID:29936722) format I believe is not a correct way to cite the paper- please add to references the proper citation.

Round 3

Reviewer 1 Report

Comments and Suggestions for Authors

Thank you for updating epidemiology